# Flexible Fiber Membrane Based on Carbon Nanotube and Polyurethane with High Thermal Conductivity

**DOI:** 10.3390/nano11102504

**Published:** 2021-09-26

**Authors:** Yuanzhou Chen, Junlin Chen, Yingming Zhang, Ziyue Hu, Weijian Wu, Xiang Chen, Zhifeng Hao

**Affiliations:** 1School of Chemical Engineering and Light Industry, Guangdong University of Technology, Guangzhou 510006, China; m15521162019@163.com (Y.C.); 2112106060@mail2.gdut.edu.cn (J.C.); zhuos510@163.com (Y.Z.); huzydmn@163.com (Z.H.); w13247361677@163.com (W.W.); 2Guangdong Provincial Key Laboratory of Plant Resources Biorefinery, Guangdong University of Technology, Guangzhou 510006, China

**Keywords:** multiwalled carbon nanotubes, electrospinning, flexible fiber membrane, thermal conductivity

## Abstract

The development of high thermally conductive polymer composites with low filler content remains challenging in the field of thermal interface materials (TIMs). Herein, we fabricated a series of flexible fiber membranes (TMMFM) with high thermally conductive based on thermoplastic polyurethane (TPU) and acidified multiwalled carbon nanotubes (a-MWCNTs) via electrospinning and ultrasonic anchoring method. The SEM and TEM results demonstrated that the a-MWCNTs aligned along the fiber orientation in the membrane and anchored on the membrane surface strongly, which can establish the heat conduction path both in the horizontal and vertical directions. With the incorporation of 10 wt% a-MWCNTs, the horizontal direction (λ_∥_) and vertical direction (λ_⊥_) thermal conductivity value of TMMFM-5 was 3.60 W/mK and 1.79 W/mK, respectively, being 18 times and 10 times higher compared to pure TPU fiber membranes. Furthermore, the TMMFM maintained favorable flexibility of the TPU matrix because the small amount of a-MWCNTs only slightly hinders the mobility of the TPU molecular chain. The performance of the obtained TMMFM unveils their potential as a promising choice of flexible TIMs.

## 1. Introduction

Thermoplastic polyurethane (TPU) is widely used in the electronic industry owing to its outstanding chemical resistance, excellent flexibility, elasticity and processability [1,2,3,4]. Therefore, TPU has great application potential in thermal interface materials (TIMs), which play a crucial role in efficient thermal management [5,6,7,8,9]. However, the intrinsic low thermal conductivity (λ) of TPU (~0.2 W/mK [10]) has restricted its further application in TIMs. In order to improve the polymer’s thermal transport property, incorporating it with inorganic high thermally conductive fillers is considered one of the most effective methods [11,12,13,14].

Due to the high thermal conductivity (~3000 W/mK [15]), thermal stability, elastic modulus and aspect ratio of one-dimensional carbon nanotubes (CNTs), they are broadly used as thermally conductive filler to improve the thermal conductivity of polymer matrices [16,17]. Zhu et al. [18] reported that the λ value increased by 71.7% in geopolymer-based composites when 5 vol% SiO_2_-CNTs was added. Bustero et al. [19] introduced 5 wt% CNTs into an epoxy resin matrix, and the λ value increased by 340% compared to that of pure epoxy resin. However, the reported λ values of CNT-based composites remain below expectations. This might be ascribed to the aggregation and disorientation of CNTs, which largely attenuate thermal conductivity in a certain direction [20,21]. Moreover, improvement in thermal conductivity was often limited due to phonon scattering when heat passed through the interface. Therefore, building a continuous thermally conductive highway in composites to reinforce the heat conduction capacity remains challenging, especially with low filler content.

Compared with common methods (CVD, cross-linking, ice-templated method, etc.), the electrospinning technique is a common and facile method for preparing oriented polymers and nanofibers [22,23,24,25,26]. Gu et al. [27] prepared thermally conductive boron nitride/polyimide (BN/PI) composites by electrospinning and hot-pressing technology. When the content of BN amounted to 30 wt%, the λ value increased from 0.174 W/mK to 0.696 W/mK. Dong et al. [28] fabricated polyacrylonitrile (PAN) by incorporating it with 2 wt% multiwalled CNTs (MWCNTs) via electrospinning, and the λ value increased by about 134% in PAN-based composites. Based on the above work, it is established that the electrospinning technique is beneficial to the directional arrangement of fillers in nanofibers, which will help to enhance in-plane heat conduction. However, out-of-plane heat conduction is limited. Furthermore, the above studies fail to accomplish significant improvement in the thermal conductivity of composites at low filler content. Hence, it is difficult to achieve high thermal conductivity of composites under low CNT content only with electrospinning technology, especially out-of-plane high thermal conductivity. It is worth noting that Guan et al. [29] obtained electrospun polyamide (PA 66) nanofiber bundles with increased mechanical property and conductivity by absorbing MWCNTs on the surface. This research testifies that absorbing filler on the surface of electrospinning fiber can greatly upgrade its performance.

Inspired by Guan’s work, herein, we studied the incorporation of a facile electrospinning technique with the ultrasonic anchoring method to fabricate thermally conductive and flexible fiber membranes based on a-MWCNTs and TPU, which greatly improved the heat conduction path both in the horizontal and vertical directions. Under the action of electrospinning, a-MWNCTs were distributed along the axial direction inside TPU fibers, resulting in an increase in the thermal conductivity of TMMFM in the horizontal direction. With the dual synergy of ultrasonication and intermolecular hydrogen bonding, a-MWNCTs could be evenly anchored on the surface of TPU fibers, which promoted the thermal conductivity of TMMFM in the vertical direction at a-MWCNT low loading. In addition, TMMFM could maintain its flexibility because the small amount of a-MWCNTs only slightly hindered the mobility of the TPU molecular chain. We believe that this methodology may be favored in guiding the future design of flexible fiber membrane composites with high thermal conductivity.

## 2. Experimental Methods

### 2.1. Materials 

The MWCNTs (average length = 3–15 μm, average diameter = 10–20 nm) were purchased from Nanjing Xianfeng Nano Material Technology Co., Ltd. Thermoplastic polyurethane (TPU) (Elastollan 1185A) with a density of 1.12 g/cm^3^ and a melt flow index of 17.5 g/10 min (215 °C, 10 kg) was supplied by BASF Co., Ltd. Sulfuric acid (H_2_SO_4_, 98%), nitric acid (HNO_3_, 70%), acetone and N, N-dimethylformamide (DMF) were purchased from Guangzhou Chemical Co., Ltd. All materials were used as received without any further purification.

### 2.2. Synthesis of Acidified MWCNTs (a-MWCNTs)

First, 1 g of MWCNTs was added to a 100 mL mixed solution of sulfuric acid and nitric acid (volumetric ratio = 3:1) and stirred at 80 °C for 8 h. The mixture was then repeatedly centrifuged and washed with deionized water until the pH was close to 7. Finally, the suspension was filtered with a cellulose acetate membrane with a diameter of 0.45 μm and dried in a 40 °C oven for 12 h to obtain a-MWCNTs.

### 2.3. Fabrication of a-MWCNTs@TPU-a-MWCNTs Fiber Membranes (TMMFM)

The TPU matrix was first dissolved in the solvent DMF, and a-MWCNTs fillers (mTPU/ma-MWCNTs = 19:1) were added and dispersed by mechanical stirring for 6 h to obtain the corresponding electrospinning solution. The prepared electrospinning solution was loaded into a plastic syringe and fed through a metallic nozzle at a feed rate of 1 mL/h. The applied voltage was 15 kV, the temperature was 25 °C and the distance between the metallic needle and the rotating drum was 15 cm. The a-MWCNTs@TPU fiber membranes were collected on aluminum foil, which was attached to the rotating drum. Afterward, a-MWCNTs@TPU fiber membranes were immersed in the a-MWCNTs dispersion (0.5 mg/mL) and ultrasonicated for 1 min, 5 min and 10 min, respectively, then dried in a 50 °C oven for 24 h to obtain a-MWCNTs@TPU-a-MWCNTs fiber membranes with additional content of approximately 1 wt%, 3 wt% and 5 wt% a-MWCNT, respectively. For comparison, TPU fiber membranes were prepared according to the same method. For convenience, TPU, a-MWCNTs@TPU and three different a-MWCNTs loading a-MWCNTs@TPU-a-MWCNTs fiber membranes were denoted as TFM, TMFM, TMMFM-1, TMMFM-3 and TMMFM-5, respectively. The detailed constitution of fiber membranes is listed in Table 1, and the preparation process of TMMFM is presented in Figure 1.

### 2.4. Characterization

Fourier transform infrared (FTIR) spectra measurements were performed with a resolution of 4 cm^−1^ and a range from 500 to 4000 cm^−1^ (Thermo Fisher Nicolet 6700, Waltham, MA, USA). The morphologies of MWCNTs, a-MWCNTs and fiber membranes were examined using a scanning electron microscope (SEM, Phenom-World ProX, Eindhoven, Netherlands) and a transmission electron microscope (TEM, HITACHI HT7700, Tokyo, Japan). MWCNTs and a-MWCNTs were configured as nearly clear and transparent dispersion in the TEM sample preparation process to avoid any unfavorable factors. Electrospun nanofibers were directly deposited onto a copper grid for a few seconds during electrospinning for TEM observation. The mechanical properties of fiber membranes were determined using a tensile strength machine with a 500 N load cell (SUST CMT-2202, Zhuhai, China) with a constant rate of 10 mm/min. Rectangular strips with dimensions of 60 × 10 × 0.25 mm^3^ were cut off from the fiber membranes for the mechanical performance test. The rectangular samples were fixed by two insulating jigs of the universal tensile testing machine, and the gage length was 20 mm. Differential scanning calorimetry (DSC) analyses of the specimens were performed at 10 °C/min (nitrogen atmosphere) over the temperature range of −50 °C to 40 °C using DSC3 (METTLER, Zurich, Switzerland). The thermal diffusivity of the samples was measured using an LFA 467 light flash system (NETZSCH, Selbu, Germany) at 25 °C. The dimensions of the samples for the cross-plane thermal diffusivity test were 100 × 100 × 0.25 mm^3^, and for the in-plane thermal diffusivity test were 254 mm in diameter and 0.25 mm in thickness. All specimens were coated with a thin layer of fine graphite powder on both sides to ensure that the samples were covered tightly without light leakage and uniform heat transfer. Thermal conductivity (λ) was calculated from the formula:$$\lambda =\alpha \times \rho \times C_{P}$$
where *α* is the thermal diffusivity of the samples; *ρ* is the density of the samples, which is calculated by the weight and volume of the samples with dimensions of 100 × 100 × 0.25 mm^3^; and *C_P_* is the specific heat capacity, which is obtained by DSC. The thermal images and the temperature of the samples were captured by an infrared camera (MAGNITY MAG12-111703, Shanghai, China) during the test. Three samples were prepared for each sample to test for all the methods.

## 3. Results and Discussions

### 3.1. Morphology

Figure 2 presents the morphologies of MWCNTs, a-MWCNTs, TFM, TMFM and TMMFM-5. It can be seen from Figure 2a that the winding between MWCNTs is very serious, owing to the large aspect ratio of MWCNTs and relatively strong van der Waals force between the tubes. After surface modification, the winding phenomenon of MWCNTs was greatly improved, as shown in Figure 2b. Figure 2b also confirms that a-MWCNTs are well dispersed compared to MWCNTs. Moreover, as shown in Appendix A, after acidification, the peaks of C=O stretching and −OH bending of MWCNTs at 3430 and 1627 cm^−1^ are significantly enhanced. All the results indicate that the surfaces of a-MWCNTs are rich in −OH and −COOH functional groups. As evidenced by the SEM image presented in Figure 2d, TPU fibers are randomly distributed with a smooth surface. However, the surface of TMFM becomes sharply rough, as shown by the red arrows in Figure 2e, with the loading of a-MWCNTs. The TEM of TMFM indicates that the protrusion of fiber is mainly ascribed to the bending and winding of the intrinsic structure of a-MWCNTs (represented by blue arrows in Figure 2c). Additionally, a-MWCNTs have a certain orientation in TPU fibers with uniform distribution and no apparent agglomeration (see yellow arrows in Figure 2c). In Figure 2f, the nanofiber is densely decorated by a-MWCNTs to form a hierarchical micronanostructure. It is worth noting that a-MWCNTs are homogeneously covered on the surface of nanofibers under the action of ultrasound, avoiding a-MWCNTs aggregation on the nanofiber surface. Furthermore, a-MWCNTs were firmly anchored on the surface of TPU fibers. The reasons are as follows: (1) under the action of ultrasonication, the a-MWCNTs collide with the fibers at high speed and remain on the fibers. (2) With strong intermolecular hydrogen bonding between a-MWNCTs and TPU, a-MWNCTs can be strongly anchored on the surface of TPU fibers.

### 3.2. Thermal Stability

TGA and DSC curves of TFM, TMFM and TMMFM are shown in Figure 3a,b, and the corresponding characteristic thermal data are presented in Table 2. As shown in Table 2, both the onset degradation temperature (*T_5_*: 5 wt% degradation) and the maximum weight loss rate temperature (*T_max_*) of TMFM are higher than that of TFM. Furthermore, the glass transition temperature (*T_g_*) value TMFM enhances from −31.9 °C to −23.1 °C. Clearly, the temperature increase of *T_5_*, *T_max_* and *T_g_* is related to the addition of MWCNTs. The existence of a small amount of a-MWCNTs might restrict the movement of the TPU molecular chains, promoting the thermal stability and rigidity of fiber membranes. On the other hand, the TPU molecular chains can be oriented during the electrospinning process, which also contributes to the increase in thermal stability and rigidity of all the fiber membranes. However, compared to that of TMFM, the *T_5_* and *T_max_* values of TMMFM-1, TMMFM-3 and TMMFM-5 slightly decrease to 317.2 °C, 405.1 °C, 297.9 °C, 397.4 °C and 297.1 °C, 394.6 °C, respectively, and the corresponding *T_g_* value decreases to −24.6 °C, −26.6 °C and −27.4 °C, respectively. The results indicate the thermal stability of TMMFM is slightly degraded after anchoring a-MWCNTs onto the surface of TPU fibers. The slightly decreased thermal stability performance may be associated with the high thermal conductivity of MWCNTs (~3000 W/mK [30]). With the action of ultrasonication, the surface of TPU fibers is covered and anchored by a-MWCNTs (shown in Figure 2f). Therefore, with the increase in thermal conductivity, heat can be efficiently transferred in the fibers, which accelerates the decomposition of TMMFM. As for the slightly decreased *T_g_* value of the TMMFM, it may be also related to the surface of fibers. After the a-MWCNTs anchored on the surface of TMFM, the oriented TPU molecular chains may be slightly damaged, leading to easy relaxation for the molecular segment [31,32].

### 3.3. Mechanical Property

To investigate the influence of the decoration of a-MWNCTs on mechanical properties, monotonic tensile tests of TFM, TMFM and TMMFM were carried out at a tensile rate of 10 mm/min. Figure 4a presents the typical stress–strain curves of TFM, TMFM and TMMFM. It is noteworthy that TMFM and TMMFM display an obvious decline in elongation at break compared to that of TFM, but the elongation at break for TMFM and TMMFM remains about 350%. Moreover, TFM and TMMFM can be easily rolled into a coil-like shape, as shown in Figure 4c, indicating that the flexibility of TMMFM can be maintained. The main chain structure of TPU plays a decisive role in the flexibility of polymer chains. A small amount of a-MWCNTs only slightly hinders the rotation of the TPU molecular chain, so TMMFM can maintain the intrinsic flexibility of the TPU matrix. Compared to TFM, the tensile strength of TMFM and TMMFM increased, especially the tensile strength of TMFM, which increased from 5.3 ± 0.7 MPa for TFM to 7.7 ± 0.8 MPa. It is reported that the inorganic fillers can act as a nucleating agent in the crystallization process of polymer composites to improve their crystallinity [33]. Therefore, the increasing tensile strength of the TMFM and TMMFM composites is denoted from the inorganic filler MWCNTs material. In addition to the reasons described above, a schematic diagram of the fracture process of TMFM is shown in Figure 4b, and the other reasons for the reinforcement of composites tensile strength are as follows [34]: (1) The orientation arrangement and slip of MWCNTs at the fracture site require additional energy. (2) The more evenly dispersed MWCNTs are in TPU, the lower the probability of stress concentration is, making the distribution of cracked area more homogeneous. (3) The surface of MWCNTs (with the adhesion promoting COOH groups) improves CNT-TPU adhesion via hydrogen bonding interaction, which then, together with the good mechanical strength of CNTs, is reflected as the good strength of the composites.

The hydrogen bonding that enhanced interfacial compatibility between TPU and MWCNTs was explored via FTIR, as shown in Figure 5. In the FTIR spectrum of TFM, the peak at 3321 cm^−1^ is the characteristic N-H stretching band of urethanes, and the peak at 2937 cm^−1^ is the adsorption peak of C-H stretching vibrations. The two strong vibrations at around 1729 cm^−1^ and 1596 cm^−1^ belong to -H-N-COO-. Moreover, the peaks at 1529 cm^−1^ and 1076 cm^−1^ belong to N-H bending and C-O-C bands, respectively. After introducing a-MWCNTs into TPU fibers, the absorption peak at 3321 cm^−1^, 2937 cm^−1^, 1729 cm^−1^, 1529 cm^−1^ and 1076 cm^−1^ shifted to 3315 cm^−1^, 2935 cm^−1^, 1728 cm^−1^, 1526 cm^−1^ and 1073 cm^−1^, respectively [31,32,35]. Such changes indicated that there was hydrogen bonding interaction between TPU molecular chains and a-MWCNTs. It is notable that the as-discussed hydrogen bonding interaction not only contributed to the tensile strength of TPU composite fiber membranes but also contributed to form a heat conduction pathway to promote the thermal conductivity performance.

### 3.4. Thermal Conductivity

To explore the correlation between the structure of the TMMFM network and the λ of the composites, the cross-plane (λ_⊥_) and in-plane (λ_∥_) thermal conductivity were studied. As can be seen in Figure 6a,b, with the mass fraction of a-MWCNTs increasing, the α and λ values of the composites increase greatly. In particular, the λ_⊥_ and λ_∥_ values of TMMFM-5 are significantly improved to 1.79 W/mK and 3.6 W/mK, about 13 and 18 times higher than that of TFM (λ_⊥_ of 0.18 W/mK and λ_∥_ of 0.2 W/mK), respectively. In order to reflect the excellent thermal conduction enhancement of MWCNTs, in this study, some previously reported CNT-based film-like composites are summarized in Figure 7 and Appendix A [19,28,36,37,38,39]. Among these studies, Jang et al. [39] reported that the λ value of MWCNT/PA6 increased by 232% when 10 wt% MWCNTs were introduced. Obviously, in our work, the enhancement of thermal conductivity for TMMFM-5 (with 10 wt% a-MWCNTs loading) is significant in comparison with similar research.

The schematic diagram of the heat conduction route of TMMFM is depicted in Figure 8. The strong hydrogen bonding interaction between a-MWCNTs and TPU and electrospinning favored a-MWCNTs oriented in TPU fibers in the horizontal direction. In addition, hydrogen bonding interaction effectively improved interfacial compatibility and reduced interfacial thermal barriers between a-MWCNTs and the TPU matrix. Consequently, the high orientation of a-MWCNTs and hydrogen bonding interaction between a-MWCNTs and TPU accelerated phonon transmission, resulting in enhancement of the λ value of the composites. Furthermore, with a-MWCNTs anchored onto the surface of TPU fibers by ultrasonic anchoring technology, a-MWCNTs interconnected to form more efficient heat conduction pathways in the vertical direction. The combination of these strategies greatly promoted the heat dissipation capacity of TPU fiber membranes.

In order to further study the heat transfer performance of the composites, we placed the fiber membranes on a heating plate with constant temperature (70 °C) and recorded their surface temperature at different times by infrared thermal imaging instrument. Figure 9a,b shows the corresponding infrared thermal images and experimental results of fiber membranes. In Figure 9b, with the increase in a-MWCNTs loading anchored on the surface of TPU fibers, the surface temperature of TMMFM increased faster, indicating higher thermal conductivity and better heat transfer capability of the composites. Furthermore, the temperature distribution on the surface of each specimen is even (see Figure 9a), which verifies that a-MWCNTs maintains a uniform dispersion on the macro level.

## 4. Conclusions

In summary, flexible and high thermal conductive TMMFM were prepared by incorporating electrospinning with ultrasonic anchoring technology. With the electrospinning process, a-MWNCTs were distributed inside the TPU fiber along the axial direction, thereby reducing the heat transfer resistance and increasing the λ_∥_ value of TMMFM. Through the dual synergistic effect of ultrasonication and intermolecular hydrogen bonds, a-MWNCTs could be uniformly anchored on the surface of TPU fibers, which increased the λ_⊥_ value of TMMFM under low loading of a-MWCNTs. With the incorporation of 10 wt% a-MWCNTs, the λ_∥_ and λ_⊥_ values of TMMFM-5 were 3.60 W/mK and 1.79 W/mK, respectively, being 18 times and 10 times higher compared to pure TPU fiber membranes. Furthermore, the TMMFM maintained favorable flexibility of the TPU matrix because a small amount of a-MWCNTs only slightly hindered the mobility of the TPU molecular chain. The above-mentioned results unveil the potential for the obtained TMMFM as a promising choice for use in flexible TIMs.

## Figures and Tables

**Figure 1 nanomaterials-11-02504-f001:**
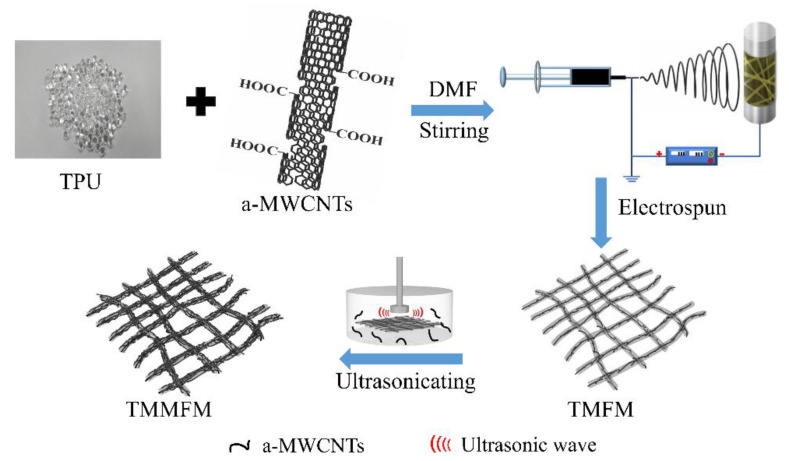
Schematic diagram of preparation of TMMFM.

**Figure 2 nanomaterials-11-02504-f002:**
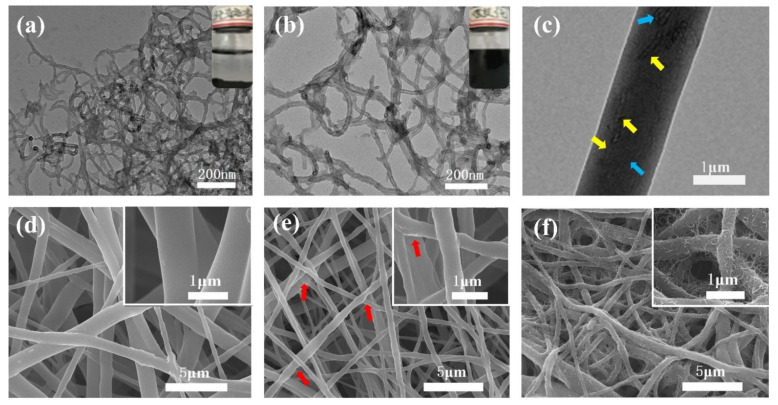
Morphologies of the composites. (**a**–**c**) TEM images of MWCNTs (**a**), a-MWCNTs (**b**) and TMFM 201 (**c**). The insets of (**a**,**b**) are MWCNTs and a-MWCNTs dispersion, respectively. (**d**–**f**) SEM images of pure TFM (**d**), TMFM 202 (**e**) and TMMFM-5 (**f**).

**Figure 3 nanomaterials-11-02504-f003:**
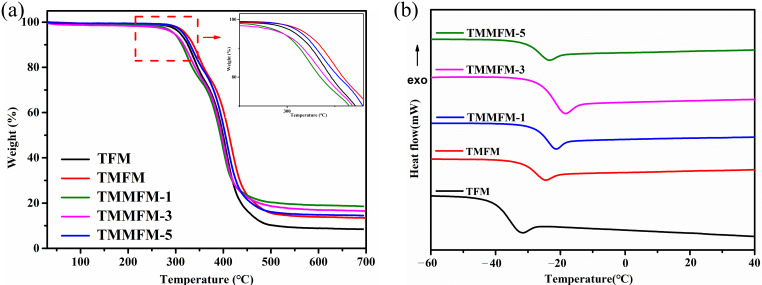
TGA (**a**) and DSC (**b**) curves of TFM, TMFM and TMMFM.

**Figure 4 nanomaterials-11-02504-f004:**
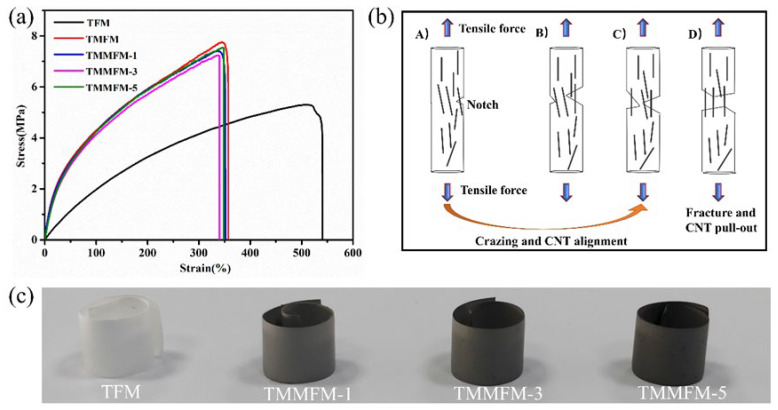
(**a**) Representative stress–strain curves of TFM, TMFM and TMMFM. (**b**) Schematic. diagram of the fracture process of TMFM. (**A**) Notches form as a result of tensile force, (**B**) notches are stretched at the crazing area, (**C**) crazing extends across the fibers with CNTs well aligned and (**D**) crazing fibers are broken. CNTs reinforce the fiber by pull-out from the TPU matrix. (**c**) Digital images of TFM and TMMFM.

**Figure 5 nanomaterials-11-02504-f005:**
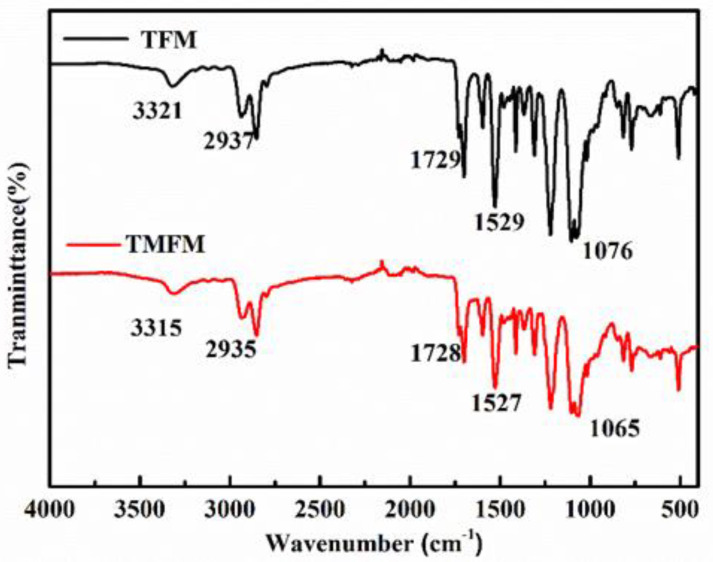
FTIR spectra of TFM and TMFM.

**Figure 6 nanomaterials-11-02504-f006:**
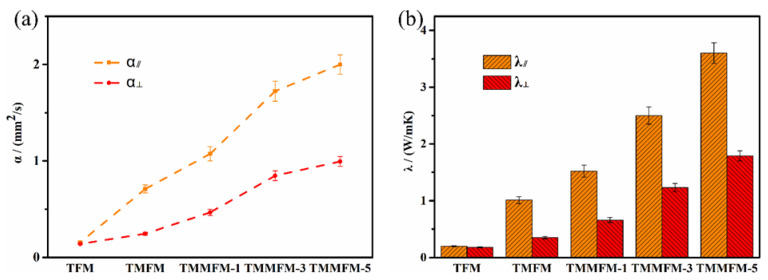
(**a**) α_⊥_ and α_∥_ values of TFM, TMFM and TMMFM. (**b**) λ_⊥_ and λ_∥_ values of TFM, TMFM and TMMFM.

**Figure 7 nanomaterials-11-02504-f007:**
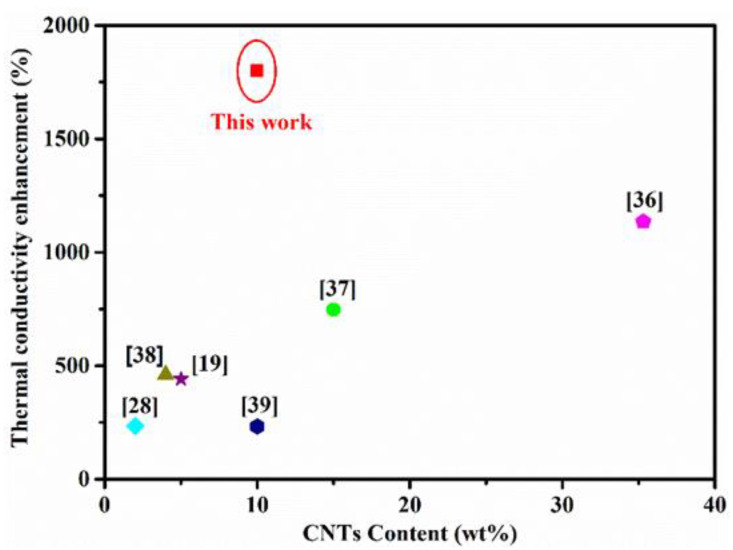
Thermal conductivity enhancement of this work comparison with others.

**Figure 8 nanomaterials-11-02504-f008:**
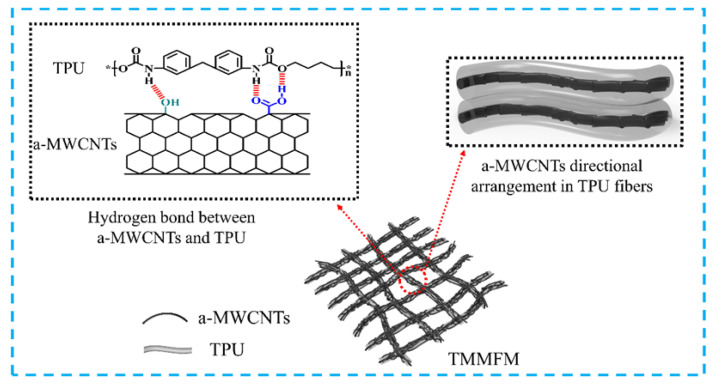
Thermal conductivity mechanical schematic diagram of TMMFM.

**Figure 9 nanomaterials-11-02504-f009:**
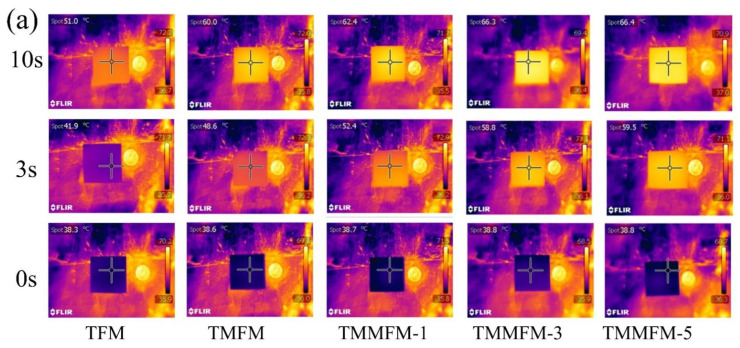
(**a**) Infrared thermal images of the TFM, TMFM and TMMFM. (**b**) Infrared thermal imaging temperature chart of TFM, TMFM and TMMFM.

**Table 1 nanomaterials-11-02504-t001:** Constituents of composite membranes.

Sample	TFM	TMFM	TMMFM-1	TMMFM-3	TMMFM-5
TPU, wt%	100	95	94	92	90
a-MWCNTs, wt%	0	5	6	8	10

**Table 2 nanomaterials-11-02504-t002:** Thermal characteristics of TFM, TMFM and TMMFM.

Sample	*T_5_* (°C)	*T_max_* (°C)	*T_g_* (°C)
TFM	309.0	394.2	−31.9
TMFM	321.4	412.8	−23.1
TMMFM-1	317.2	405.1	−24.6
TMMFM-3	297.9	397.4	−26.6
TMMFM-5	297.1	394.6	−27.4

## Data Availability

Data are contained within the article.

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
