# Peer review of "Flexible Fiber Membrane Based on Carbon Nanotube and Polyurethane with High Thermal Conductivity"

_nanomaterials, 2021, doi:10.3390/nano11102504_

Round 1

Reviewer 1 Report

Basically, the article is clear and the results are good and interesting. However, there are some missing information and points to be improved before publishing it.

As the used materials are commercial, it would be nice to have more exact product names for MWCNTs and TPU.

How the wt% of MWCNTs in the fibres was verified?

The method to produce the fibrous films should be described somewhere. Now, only the materials and the electrospinning are described but not the fiber film formation.

Chapter 2.4 should be more complete on describing details about tensile test (sample dimension, load cell size), density measurement method for the LFA and number of parallel samples for all methods. Also the TEM sample preparation method should be described as the entanglement of the samples is analysed and it can be affected by the sample preparation method. If a solvent is used to prepare the samples, does the different solubility of MWCNT and a-MWCNT affect the analysis?

Some arrows or other indicators could be used in fig. 2c to support the conclusions described on pages 144-146.

Row 175: I think it shoudl be the ’the maximum weight loss rate temperature’ not ’the rate of maximum weight loss temperature’. How this was defined, from the maximum value of DTG curve (this should be described)?

Row 176: it is stated that the Tg is ’enhanced’ (a positively loaded word) as it increases towards higher temperature. Considering the application, wouldn’t it be more beneficial if Tg would be as low as possible to keep the polymer in its rubbery state throughout the possible operation temperatures?

On row 180 it is stated that ‘On the other hand, the TPU molecular chains can be oriented during the electrospinning process, which also contributed to the increasing of thermal stability and rigidity of fiber membranes’ but if here the TMFM and TFM are compared, isn’t both of them electrospinned?

It would be nice to have also the onset degradation temperature in table 2 as they are highlighted in Figure 3.

Row 225: The text is not clear on authors idea how crystallinity affects in the case of TPU-based composites. It is introduced but then not considered in figure 4b. Further, in Figure 4b the A), B) etc steps should be described in the text.

Row 232: It is stated that ‘Due to the outstanding mechanical properties of MWCNTs, the tension can be effectively transferred to the compatible TPU-MWCNTs interface which was enhanced with the hydrogen bonding.’ I think the causality of the sentence is wrong. It is the surface of the MWCNT (with the adhesion promoting COOH groups) that improves the CNT-TPU adhesion which then together with good mechanical strength of CNTs is reflected as good strength of the composite. The good mechanical properties of CNTs is not the reason for good adhesion.

Row 340. The details for the infrared camera system and its calibration should be given in methodology section.

A space should be added between values and units (eg row 102)

Although the language is rather good, the use of senses (past/present) is here and there peculiar and against scientific norm.

Author Response

We would like to express our gratitude towards the reviewers regarding the comments. At the same time, we also like to thank the editor for the time and efforts paid. Accordingly, we made respective revisions in the manuscript. In the following parts, we had pointed out the revisions in the manuscript and try our best to reply to the reviewers’ comments.

  1. As the used materials are commercial, it would be nice to have more exact product names for MWCNTs and TPU.

Thank you for this suggestion. The MWCNTs (average length = 3~15 μm, average diameter = 10~20 nm) were purchased from Nanjing Xianfeng Nano Material Technology Co., Ltd. Thermoplastic polyurethane (TPU) (Elastollan 1185A) with a density of 1.12 g/cm3 and a melt flow index of 17.5 g/10 min (215 ºC, 10 kg) was supplied by BASF Co., Ltd. (Revised manuscript, Row 86-89)

  1. How the wt% of MWCNTs in the fibers was verified?

In TMFM, the mass ratio of TPU to MWCNTs is 19:1, so the mass fraction of MWCNTs in TMFM is 5 wt%. TMMFM were obtained by ultrasonicating TMFM in a-MWCNTs dispersion for different times. By controlling the ultrasonic time, the mass fraction of a-MWCNTs (1 wt%, 3wt% and 5wt%) anchored on TMFM can be controlled. After drying, TMMFM were weighed and calculated to obtain the mass fraction of MWCNTs. (Revised manuscript, Row 100-101)

3.The method to produce the fibrous films should be described somewhere. Now, only the materials and the electrospinning are described but not the fiber film formation.

The preparation method of fiber membranes was firstly provided in the supplementary material. And we rearranged this section to main text to emphasize the method to produce the fibrous films as the reviewer suggested. (Revised manuscript, Row 100-106)

  1. Chapter 2.4 should be more complete on describing details about tensile test (sample dimension, load cell size), density measurement method for the LFA and number of parallel samples for all methods. Also the TEM sample preparation method should be described as the entanglement of the samples is analysed and it can be affected by the sample preparation method. If a solvent is used to prepare the samples, does the different solubility of MWCNT and a-MWCNT affect the analysis?

 (1) We have described the details about TEM sample preparation method, tensile test, density measurement method and number of parallel samples in Chapter 2.4 according to the reviewer's suggestion. (Revised manuscript, Row 129-132, 133-138, 149-151, 153-154)

(2) The a-MWCNTs have better dispersion in water, but we prepared MWCNTs and a-MWCNTs into nearly clear and transparent dispersion in the TEM sample preparation process to avoid any unfavorable factors.

  1. Some arrows or other indicators could be used in fig. 2c to support the conclusions described on pages 144-146.

We have added yellow and blue arrows to Fig. 2c to support the conclusions. (Revised manuscript, Figure 2c, Row 168-171)

  1. Row 175: I think it should be the ’the maximum weight loss rate temperature’ not ’the rate of maximum weight loss temperature’. How this was defined, from the maximum value of DTG curve (this should be described)?

(1) We have placed ‘the maximum weight loss rate temperature’ with ‘the rate of maximum weight loss temperature’ in the text. (Revised manuscript, Row 209)

(2) We have listed the maximum weight loss rate temperature (Tmax: peak temperature from the DTG curves) in Table 2. In order to better typesetting, we have not placed the DTG diagram in the text. (Revised manuscript, Table 2, Row 249-250)

  1. Row 176: it is stated that the Tg is ’enhanced’ (a positively loaded word) as it increases towards higher temperature. Considering the application, wouldn’t it be more beneficial if Tg would be as low as possible to keep the polymer in its rubbery state throughout the possible operation temperatures?

During the electrospinning process, a-MWCNTs were embedded in the TPU fibers, which would limit the movement of TPU molecular chains, and the Tg value of TMFM and TMMFM will increase slightly compared to that of TFM according to the results. But more importantly, the existence of a-MWCNTs can greatly improve the heat conduction performance of fiber membranes.

  1. On row 180 it is stated that ‘On the other hand, the TPU molecular chains can be oriented during the electrospinning process, which also contributed to the increasing of thermal stability and rigidity of fiber membranes’ but if here the TMFM and TFM are compared, isn’t both of them electrospinned?

Both of TFM and TMFM were prepared via electrospinning technique which were helpful to increase their thermal stability. The sentence ‘On the other hand……rigidity of fiber membranes’ only wants to explain why the thermal stability of TMFM was improved, rather than compare with that of TFM. We have placed the sentence ‘On the other hand……fiber membranes’ with the sentence ‘On the other hand …… all the fiber membranes’. (Revised manuscript, Row 214-217)

  1. It would be nice to have also the onset degradation temperature in table 2 as they are highlighted in Figure 3.

We have added the onset degradation temperature (T5: 5 wt% degradation occurs) of fiber membranes in Table 2. (Revised manuscript, Table 2, Row 249-250)

  1. Row 225: The text is not clear on authors idea how crystallinity affects in the case of TPU-based composites. It is introduced but then not considered in figure 4b. Further, in Figure 4b the A), B) etc steps should be described in the text.

 (1) The sentence ‘It is reported that …… to improve its crystallinity’ in the text is one of the reasons why the tensile strength of TMFM would increase. And Figure 4b discusses the other reasons for the reinforcement of TMFM tensile strength in addition to the reason mentioned above. (Revised manuscript, Row 263-269)

(2) We have added the A), B) etc. steps in the text. (Revised manuscript, Fig. 4b, Row 310-312)

11.Row 232: It is stated that ‘Due to the outstanding mechanical properties of MWCNTs, the tension can be effectively transferred to the compatible TPU-MWCNTs interface which was enhanced with the hydrogen bonding.’ I think the causality of the sentence is wrong. It is the surface of the MWCNT (with the adhesion promoting COOH groups) that improves the CNT-TPU adhesion which then together with good mechanical strength of CNTs is reflected as good strength of the composite. The good mechanical properties of CNTs is not the reason for good adhesion.

We have revised the original expression in the text according to the reviewer's suggestion. We have placed the sentence ‘Due to …… hydrogen bonding’ with the sentence ‘the surface of the MWCNT…… the composite’ (Revised manuscript, Row 272-275)

  1. Row 340. The details for the infrared camera system and its calibration should be given in methodology section.

We have added the model for the infrared camera system in Chapter 2.4. And we have consulted the manufacturer, and they said that the temperature calibration has been done before leaving the factory and during installation by the engineer. (Revised manuscript, Row 151-153)

  1. A space should be added between values and units (eg. row 102)

We have added space between values and units in the text. (Revised manuscript, Row 54, 112)

  1. Although the language is rather good, the use of senses (past/present) is here and there peculiar and against scientific norm.

We have carefully reviewed and revised the manuscript and try our best to corrected all the grammatical errors that have been found.

Reviewer 2 Report

Manuscript: Nanomaterials-1382738

Flexible fiber membrane based on carbon nanotube and polyurethane with high thermal conductivity”

by Y. Chen et al

The manuscript describes experimental work on flexible fiber membranes based on thermoplastic polyurethane (TPU) and acidified multi-walled carbon nanotubes (a-MWCNTs) as filler, prepared via electrospinning method (TMFM) and ultrasonic anchoring of additional amount of a-MWCNTs (TMMFM-x, x=1,3,5). The manuscript focus on the enhanced thermal conductivity of the membranes.  The experiments are well designed and carefully performed and the experimental results are interesting. The presented results will be of interest for the corresponding scientific community and I suggest the acceptance of the manuscript for publication in Nanomaterials.

However, the following issues should be addressed before publication:

  1. Details on the measurement of cross-plane and in-plane thermal diffusivity shall be provided. The authors should also discuss the impact of the small thickness of the specimens (~ 250 μm) in the uncertainty of the measurements.
  2. What is the durability (stability in time) of the TMMFM membranes? The authors are encouraged to discuss the mechanism of a-MWCNTs anchoring on the surface of the fibers.

Author Response

We would like to express our gratitude towards the reviewers regarding the comments. At the same time, we also like to thank the editor for the time and efforts paid. Accordingly, we made respective revisions in the manuscript. In the following parts, we had pointed out the revisions in the manuscript and try our best to reply to the reviewers’ comments.

  1. Details on the measurement of cross-plane and in-plane thermal diffusivity shall be provided. The authors should also discuss the impact of the small thickness of the specimens (~ 250 μm) in the uncertainty of the measurements.

(1) We have added the detailed measurement of cross-plane and in-plane thermal diffusivity in Chapter 2.4. (Revised manuscript, Row 141-144)

(2) We will spray graphite on the surface of the fiber membranes to ensure that the surface is covered tightly without light leakage and uniform heat transfer. Moreover, three samples were made for each sample to test. And the mentioned above have been added in Chapter 2.4. (Revised manuscript, Row 144-146, 153-154)

  1. What is the durability (stability in time) of the TMMFM membranes? The authors are encouraged to discuss the mechanism of a-MWCNTs anchoring on the surface of the fibers.

(1)  The FTIR spectrum of freshly made TMMFM-5 is almost the same as that of TMMFM-5 placed under atmosphere for over a month, suggested the TMMFM is stable without any degradation.

(2) There are two main reasons why a-MWCNTs can be firmly anchored in fibers: firstly, under the action of ultrasonication, the a-MWCNTs collide with the fibers at high speed and remains on the fibers. Secondly, with strong intermolecular hydrogen bonding between a-MWNCTs and TPU, a-MWNCTs can be strongly anchored on the surface of TPU fibers. We have added the above content to Chapter 3.1. (Revised manuscript, Row 175-179) 

Reviewer 3 Report

This paper is interesting however before the final decission comprehasive major revision should be performed by authors.

My detailed comments in the attachment.

Author Response

We would like to express our gratitude towards the reviewers regarding the comments. At the same time, we also like to thank the editor for the time and efforts paid. Accordingly, we made respective revisions in the manuscript. In the following parts, we had pointed out the revisions in the manuscript and try our best to reply to the reviewers’ comments.

  1. in text 1-5 wt% was mentioned but in Table contnet is from 5-10 wt %.

Please comment on that.

In TMFM, the mass ratio of TPU to a-MWCNTs is 19:1, so the mass fraction of a-MWCNTs in TMFM is 5 wt%. TMMFM were obtained by ultrasonicating TMFM in a-MWCNTs dispersion for different times. By controlling the ultrasonic time, the mass fraction of a-MWCNTs (1 wt%, 3 wt% and 5 wt%) anchored on TMFM can be controlled. After drying, TMMFM were weighed and calculated to obtain the mass fraction of a-MWCNTs. (Revised manuscript, Row 100-101)

  1. in supplementary material please mention about FTIR spectra measurement conditions. Did authors used ATR with diamond crystal?

We have supplemented FTIR spectra measurement conditions in supplementary material. The FTIR spectra of MWCNTs and a-MWCNTs were recorded via KBr pressed disc technique, and the FTIR spectra of fiber membranes were recorded by ATR with diamond crystal.

  1. tensile test for pure TPU is rather low (only 5 MPa), any comment on that.

The tensile strength of TPU is related to molecular weight, crosslinking degree, crystallinity, etc. Liu et al. [1] obtained pure TPU membranes by combination of co-coagulation and the compression molding technique, with tensile strength about 2.5 MPa. Ren et al. [2] fabricated two pure TPU fibrous mats by electrospinning, which were named as P-TPU and V-TPU (TPU fibers parallel and vertical to the rotation direction of the drum are defined as P-TPU and V-TPU fibers, respectively.). And the tensile strength of P-TPU and V-TPU were 7.6 MPa and 0.6 MPa, respectively. Wang et al. [3] reported that the tensile strength of pure TPU is about 6 MPa via electrospinning technique. Compare with mentioned above work, the tensile strength of our TPU fiber membranes is in a reasonable and acceptable range.

  1. Liu, H.; Li, Y.; Dai, K.; Zheng, G.; Liu, C.; Shen, C.; Yan, X.; Guo, J.Guo, Z., Electrically conductive thermoplastic elastomer nanocomposites at ultralow graphene loading levels for strain sensor applications. JOURNAL OF MATERIALS CHEMISTRY C 2016, 4, (1), 157-166, https://doi.org/10.1039/c5tc02751a.
  2. Wang, Y.; Hao, J.; Huang, Z.; Zheng, G.; Dai, K.; Liu, C.Shen, C., Flexible electrically resistive-type strain sensors based on reduced graphene oxide-decorated electrospun polymer fibrous mats for human motion monitoring. Carbon 2018, 126, 360-371, https://doi.org/10.1016/j.carbon.2017.10.034.
  3. Ren, M.; Zhou, Y.; Wang, Y.; Zheng, G.; Dai, K.; Liu, C.Shen, C., Highly stretchable and durable strain sensor based on carbon nanotubes decorated thermoplastic polyurethane fibrous network with aligned wavelike structure. CHEMICAL ENGINEERING JOURNAL 2019, 360, 762-777, https://doi.org/10.1016/j.cej.2018.12.025.

Round 2

Reviewer 1 Report

A new version of the revised manuscript pdf file should be sumitted, the current one is missing Figure 2.

Author Response

We would like to express our gratitude towards the reviewers regarding the comments. Accordingly, we made respective revisions in the manuscript. In the following parts, we had pointed out the revisions in the manuscript and try our best to reply to the reviewers’ comments.

Question: A new version of the revised manuscript pdf file should be sumitted, the current one is missing Figure 2.

Answer: Figure 2 has been added to the PDF file, which may be an oversight of file conversion. The latest PDF file has been reprocessed and uploaded.
